# How to Combine Variational Bayesian Networks in Federated Learning

**Atahan Ozer**
Computer and Informatics
Istanbul Technical University
ozera17@itu.edu.tr

**Kadir Burak Buldu**
Computer and Informatics
Istanbul Technical University
buldu19@itu.edu.tr

**Abdullah Akgül**
Computer and Informatics
Istanbul Technical University
akgula15@itu.edu.tr

**Gozde Unal**
Computer and Informatics
Istanbul Technical University
gozde.unal@itu.edu.tr

## Abstract

Federated Learning enables multiple data centers to train a central model collaboratively without exposing any confidential data. Even though deterministic models are capable of performing high prediction accuracy, their lack of calibration and capability to quantify uncertainty is problematic for safety-critical applications. Different from deterministic models, probabilistic models such as Bayesian neural networks are relatively well-calibrated and able to quantify uncertainty alongside their competitive prediction accuracy. Both of the approaches appear in the federated learning framework; however, the aggregation scheme of deterministic models cannot be directly applied to probabilistic models since weights correspond to distributions instead of point estimates. In this work, we study the effects of various aggregation schemes for variational Bayesian neural networks. With empirical results on three image classification datasets, we observe that the degree of spread for an aggregated distribution is a significant factor in the learning process. Hence, we present an *survey* on the question of how to combine variational Bayesian networks in federated learning, while providing computer vision classification benchmarks for different aggregation settings.

## 1   Introduction

Over the last years, machine learning (ML) has become the de facto approach for many real-life applications. Although the long-established ML algorithms provide promising results, their requirement for central data storage in model training raises data privacy issues. In an ideal scenario, the raw data of the users should not be transferred to any external computational device to enable data-privacy considering General Data Protection Regulation [31]. However, the current state of the ML contradicts with privacy concerns. In order to address this contradiction, a Federated Learning (FL) framework, where models are being learned in a distributed manner without exposing any data to the outside, was proposed [24]. It lays the foundations of the Horizontal FL problem where users share the same feature set but a different sample space. The main objective of the FL framework is to assemble the global optimization problem by solving the local optimization problems iteratively rather than solving the global problem directly.

The structure of Horizontal FL mainly consists of two repeating stages: first, locally trained models are aggregated to construct the global model in the server; second, the global model from the previous

Workshop on Federated Learning: Recent Advances and New Challenges, in Conjunction with NeurIPS 2022 (FL-NeurIPS'22). This workshop does not have official proceedings and this paper is non-archival.

stage is distributed from the server to clients and these clients perform the local training. The first proposed algorithm with empirical results for this structure is FEDAVG [24]. It compares favorably to centralized models under the assumption that all clients have independent and identically distributed (IID) datasets. However, dataset homogeneity often is not possible in real life. In case the IID condition is not satisfied, both the convergence and the accuracy of the algorithm significantly degenerates [21, 34]. As addressed by several studies [14, 20], the performance cut of FEDAVG essentially stems from its averaging scheme. Even though the aforementioned methods offer solutions to this problem with gradient correction terms and weight space regularizations, they lack mechanisms for quantifying the uncertainty of the given predictions which is essential for safety-critical applications.

From a probabilistic perspective, another solution to this problem could be the aggregation of the distributions of the parameter space that represents the local optimization problem. This approach would provide the necessary tools for uncertainty quantification; however, the exact posterior of Deep Neural Networks are intractable due to their complexity. To resolve this issue, one could benefit from Variational Inference (VI) [3, 12] or Monte Carlo methods [9, 2, 5]. Having the required architecture for uncertainty, our motivation for this study is the question of how to aggregate the local posterior distributions to obtain a global posterior distribution.

The previous question is well studied by statisticians in the context of combining expert views on the predictive modeling of an event such as seismic risks or meteorological forecasts [6]. However, constraints and requirements for FL are remarkably different from those studies, especially for the non-IID case. In the probabilistic FL setting, there is no prior work that investigates different statistical aggregation rules for Variational Bayesian Neural Networks (VBNN) with a simple application of VI. Yet, aggregation of the clients is a significant subject of the probabilistic FL since the aggregation of clients' distributions change the outcome of the global optimization problem. In Figure 1, we illustrate that aggregation rules yield different distributions, which is exemplified with two clients.

We summarize our contributions: *i)* We empirically inspect five different statistical aggregation schemes as a *survey* for VBNNs in the federated learning setting on image classification benchmarks. *ii)* We explore VBNNs' degree of spread in the context of different federated learning scenarios with comprehensive experiments. *iii)* We examine deterministic and probabilistic models based on prediction accuracy, model calibration, and uncertainty quantification. *iv)* We share our multi-process simulation pipeline to facilitate efficient experimentation in federated learning research available at [1].

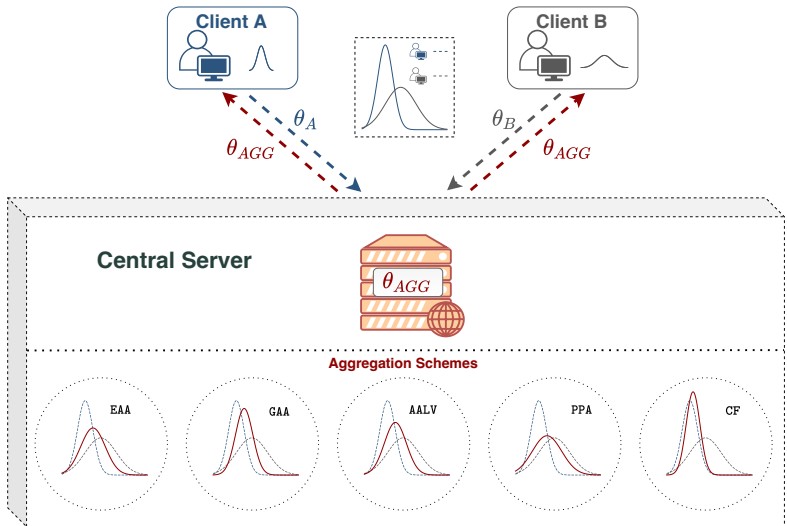

Figure 1: Different aggregation methods are depicted with two different Gaussians. First, clients obtain the weight distributions and transfer them to the central server; then the server aggregates the different distributions and returns the aggregated distribution back to the clients. The aggregation rule significantly affects the aggregated distribution. Different aggregations are described in Section 3.

---

[1] https://github.com/ituvisionlab/BFL-P

## 2 Background

### 2.1 Federated Learning

Generally, horizontal federated learning can be expressed as an assembly of $K$ local optimization problems with the following formulation:

$$\min_{\theta} f(\theta) \quad \text{where} \quad f(\theta) = \sum_{k=1}^{K} \beta_k f_k(\theta). \tag{1}$$

Depending on the machine learning problem, $f_k(\theta)$ usually corresponds to $\mathcal{L}_{\mathcal{D}_k}(\theta)$, which is the local loss function for each client $k = 1, ..., K$ utilizing its local dataset $\mathcal{D}_k$ with the model parameters $\theta$. $\beta_k$ refers to aggregation weights for $K$ local optimization problems where $\sum_{k=1}^{K} \beta_k = 1$ and $\beta_k \in (0, 1]$. Most of the time, the local stage is optimized by Stochastic Gradient Descent and its variants; nevertheless, optimizing the local problem solely is not enough for the solution of Eq. 1. To further optimize $f(\theta)$, an alternating optimization is used. First, the global model is trained for $E$ epochs on the clients with the local dataset $D_k$, and later, these locally updated models are collected in the server to apply the aggregation rule for the acquisition of the global model. After the aggregation, the new global model is distributed to the clients and the algorithm is repeated for $T$ communication rounds until convergence. In our study, $\mathcal{D}$ comprises $\mathcal{D}_k$ as one of its unique subsets: $\left\{ (x_i, y_i) \right\}_{i=1}^{|\mathcal{D}_K|}$ where $x$ refers to the features and $y$ refers to the target label for a classification problem.

One of the promising properties of the FL framework is to work with a high number of clients yet, in practice, involving all of the clients in the same communication round is time-costly and not always possible due to client inactivity. In order to simulate that effect, the whole framework is run with only $K = C \times \gamma$ active clients where $C$ is the total number of clients and $\gamma$ is a fraction for active client selection [24]. An algorithmic overview of the FL framework is given in Figure 2.

### 2.2 Variational Bayesian Neural Networks

Bayesian Neural Networks (BNN) are built on Neural Networks (NN) with a probabilistic Bayesian inference mechanism that allows them to learn the NN weights as a probability distribution instead of point estimate weights [18, 30, 7]. Using the Bayesian paradigm offers some relevant and useful outcomes such as quantification of uncertainty [16, 28], and mathematical understanding of regularizations in Deep NNs [29] which relieve the overfitting problem in NNs.

---

**Algorithm 1:** Federated Optimization

**Input:** Dataset $D$, initial $\theta$, # parties $K$, # communication rounds $T$, # local epochs $E$, learning rate $\eta$, aggregation method AGG.

**for** *each round* $t = 1, \cdots, T$ **do**
  Sample a set of parties $C_K$
  Set $C_\theta = \{\}$
  **for** $k \in C_K$ *in parallel* **do**
    $\theta_k \leftarrow$ **Client Update**$(\theta, D_k)$
    $C_\theta = C_\theta \cup \theta_k$
  $\theta \leftarrow$ AGG$(C_\theta)$ ▷ using Eqs. 4, 5, 6, 7, 8.

**Output:** $\theta$

---

**Algorithm 2:** Client Update

**Input:** Initial $\theta_k^0$, dataset $D_k$.
**for** *each epoch* $e = 1, \cdots, E$ **do**
  $\theta_k^e \leftarrow \theta_k^{e-1} - \eta \nabla \mathcal{L}_{D_k}(\theta_k^{e-1})$

**Output:** $\theta_k^E$

---

Figure 2: Federated Learning: Algorithmic Overview.

For a given classification dataset $\mathcal{D}$, BNN can be represented as a probabilistic model through its predictive density $p(y|x, \theta)$. The likelihood of the prediction can be written as $p(\mathcal{D}|\theta) = \prod_{i=1}^{|\mathcal{D}|} p(y_i|x_i, \theta)$. From Bayes' Rule, the posterior is given by $p(\theta|\mathcal{D}) = \frac{p(\mathcal{D}|\theta)p(\theta)}{p(\mathcal{D})}$. Since $p(\mathcal{D})$ (evidence) is not dependent on the model weights $\theta$, multiplication of the likelihood and $p(\theta)$ (prior of the weights) will be proportional to the true posterior of the model weights $p(\theta|\mathcal{D}) \propto p(\mathcal{D}|\theta)p(\theta)$. For complex models such as Deep NNs, an analytical solution for the true posterior is intractable. Yet, there are extensive works for approximation of the true posterior with Markov Chain Monte Carlo methods [9] or Variational Inference (VI) [3]. Its computational simplicity and scalability [4, 13] compared to the other methods, make the latter framework i.e., the Variational Bayesian Neural Networks (VBNNs) suitable for the FL problem where sources are scarce and time is limited.

Using VI, the true posterior of the model $p(\theta|\mathcal{D})$ can be approximated with a variational distribution $q_\psi(\theta)$ which is parameterized under $\psi$. Then the optimization objective $\mathcal{L}_D$ can be written as

$$\mathcal{L}_D = \mathbb{KL}\big(q_\psi(\theta)||p(\theta)\big) - \mathbb{E}_{\theta \sim q_\psi(\theta)}\big[\log p(\mathcal{D}|\theta)\big] \tag{2}$$

where $\mathbb{KL}(\cdot||\cdot)$ stands for the Kullback-Leibler (KL) divergence between the two distributions on its arguments. In this case, the optimization objective $\mathcal{L}_D$ is the negative variational free energy (Eq. 2) which corresponds to Evidence Lower Bound.

## 2.3 Federated Variational Bayesian Learning

As for many safety-critical real-world applications, BNNs are suitable to be employed in the federated learning framework. In this setup, each local optimization problem aims to approximate a local posterior $p(\theta_k|\mathcal{D}_k)$ where $k \in [1, \cdots, K]$. The problem is to minimize the $\mathcal{L}_{D_k}$ with respect to $\theta_k$, which is the weight distribution of the $k^{th}$ problem. Its prior $p(\theta_k)$ is approximated by $q_{\psi_k}(\theta_k)$ that is parameterized under $\psi_k$. Then the corresponding optimization objective reads

$$\mathcal{L}_{D_k} = \mathbb{KL}\big(q_{\psi_k}(\theta_k)||p(\theta)\big) - \mathbb{E}_{\theta_k \sim q_{\psi_k}(\theta_k)}\big[\log p(\mathcal{D}_k|\theta_k)\big]. \tag{3}$$

In practice it is hard to come up with a good prior representing the data; however, motivated by the FedProx and common usage in VAEs, we can benefit from the same gaussian priors as in regulating term. After the local optimization of each client is finished for a round, the local parameters of clients must be transferred to the server in order to obtain parameters of the global model that aims to optimize the overall loss $\mathcal{L}_D$.

The global aggregation scheme of VBNNs is different from FEDAVG and its variants since the weights of the VBNNs are not point estimates but rather they are distributions. There are several works that concern the aggregation in a probabilistic paradigm. FedSparse [23] considers the aggregation as the M-step of an Expectation-Maximization (EM) framework. pFedBayes [33] creates a personalized framework for federated Bayesian variational inference. Federated posterior averaging [1] proposes to estimate the global posterior from the clients via Monte Carlo methods. Federated online Laplace approximation [22] addresses the aggregation error and devises the usage of the Gaussian product method with Laplace approximation for averaging weights of clients. However, except FedSparse and pFedBayes, none of the aforementioned methods do use VI for Bayesian inference. Even though FedSparse and pFedBayes use VI, their main concerns are not about aggregation schemes. To our knowledge, we are the first to investigate the effects of the aggregation rule on VBNNs considering parametric distributions.

## 3 Aggregation Methods for Variational BNNs

In this section, we discuss five methods of parametric distribution aggregation for VBNNs. We introduce aggregation rules for hyperparameters of the distribution which is exemplified by a Gaussian model, where hyperparameters are mean and variance. The five methods are Empirical Arithmetic Aggregation (EAA), Gaussian Arithmetic Aggregation (GAA), Arithmetic Aggregation with Log Variance (AALV), Population Pooling Based Aggregation (PPA), and Conflation Aggregation (CF). In our work, due to its performance and accessibility; we benefit from the univariate Gaussian distribution for each neuron. Thus, we derive all aggregation methods in terms of mean ($\mu$) and variance ($\sigma^2$) parameters. As an implementation trick, since $\sigma^2 \geq 0$, we use $\alpha = \log \sigma^2$.

### 3.1 Empirical Arithmetic Aggregation (EAA)

Through the lens of Federated averaging, the most straightforward way of aggregation is to average the hyper-parameters of the client weights. If the statistical properties of Gaussians are put aside, a straightforward naive averaging yields

$$\mu_{EAA} = \sum_{k=1}^{K} \beta_k \mu_k, \quad \sigma_{EAA}^2 = \sum_{k=1}^{K} \beta_k \sigma^2. \tag{4}$$

Intuitively, aggregation with EAA is the weighted sum of clients' hyper-parameters that are $\mu$ and $\sigma^2$.

### 3.2 Gaussian Arithmetic Aggregation (GAA)

Unlike the previous approach, regarding the rigorous properties of the Gaussian, and assuming that the distribution weights of each client are mutually independent, we can use the sum rule of Gaussian

distributions to derive the following aggregation rule

$$\mu_{GAA} = \sum_{k=1}^{K} \beta_k \mu_k, \quad \sigma^2_{GAA} = \sum_{k=1}^{K} \beta_k^2 \sigma^2. \tag{5}$$

One observation about the newly obtained variance $\sigma^2_{GAA}$ of this approach is that it will be always smaller than the variance aggregated with EAA if clients provide the same variances for the two methods. The proof is straightforward, $\beta_k \in (0, 1]$ and $\beta_k^2 < \beta_k$; therefore, $\sigma^2_{GAA} < \sigma^2_{EAA}$.

### 3.3 Arithmetic Aggregation with Log Variance (AALV)

This aggregation method is derived from the implementation trick $\alpha = \log \sigma^2$ which accommodates learning $\sigma^2$ in the proper range. The aggregation method corresponds to FEDAVG and it is also used in pFedBayes, which is equivalently using gradient averaging for $\alpha$ with aggregation weights. The corresponding aggregation rule reads

$$\mu_{AALV} = \sum_{k=1}^{K} \beta_k \mu_k, \qquad \alpha_{AALV} = \sum_{k=1}^{K} \beta_k \alpha_k \implies \sigma^2_{AALV} = e^{\sum_{k=1}^{K} \beta_k \log \sigma_k^2}. \tag{6}$$

### 3.4 Population Pooling based Aggregation (PPA)

As widely studied by statisticians, there exist a variety of population pooling methods like [27]. Here, we take the most basic approach for aggregation. First, we create a set for each client by sampling from their weight distribution as $S_k = \bigcup_{i=1}^{N \cdot \beta_k} X_i \sim \mathcal{N}(\mu_k, \sigma_k^2)$ where $N$ is the population size, and $X_i$ is a sample from the weight distribution of client $k$. Afterwards, we generate a population $S_p$ from populations of all clients in order to represent local characteristics in the aggregated distribution. The $S_p$ population can be obtained with $S_p = \bigcup_{k=1}^{K} \bigcup_{i=1}^{N \cdot \beta_k} X_i$ where $X_i \in S_k$. Having defined the population for aggregation, we can obtain the hyper-parameters of the aggregated distribution by

$$\mu_{PPA} = \bar{X}_N = \frac{1}{N} \sum_{i=1}^{N} X_i, \qquad \sigma^2_{PPA} = \frac{1}{N} \sum_{i=1}^{N} (X_i - \bar{X}_N)^2, \qquad X_i \in S_p. \tag{7}$$

### 3.5 Conflation Aggregation (CF)

Conflation [10] is introduced as a unification of a finite number of probability distributions into a single probability distribution. Conflation is defined as $f(x) = \dfrac{f_1(x) f_2(x) \dots f_K(x)}{\displaystyle\int_{-\infty}^{\infty} f_1(y) f_2(y) \dots f_K(y) dy}$ where $f_k$, $k = 1, ..., K$, is a set of probability density functions to be consolidated. Conflation can be applied to any kind of probability distribution and is easy to calculate. Furthermore, conflation appears in many real-life scenarios such as [11, 35, 8] in order to fuse different measurements of the same quantity.

For a set of $K$ Gaussian distributions, the conflation aggregation equations are derived as follows:

$$\mu_{CF} = \frac{\frac{\beta_1 \mu_1}{\sigma_1^2} + \dots + \frac{\beta_k \mu_k}{\sigma_k^2}}{\frac{\beta_1}{\sigma_1^2} + \dots + \frac{\beta_K}{\sigma_K^2}}, \qquad \sigma^2_{CF} = \frac{\beta_{max}}{\frac{\beta_1}{\sigma_1^2} + \dots + \frac{\beta_K}{\sigma_K^2}}, \tag{8}$$

where $\beta_{max} = \max \beta_k$.

Conflation tends to decrease the variance of the aggregated distribution. Furthermore, conflation yields a mean value that is proportional to aggregation weights and inversely proportional to variances.

## 4 Results and Discussion

**Experiment results with 10 clients**  As it can be observed, VBNNs with a relatively low aggregation degree of spread (GAA, AALV, and CF) outperform deterministic counterparts (FED and FEDAVG) on all metrics (See Appendix 6.1 Table 4 for variance comparisons). When predictive accuracies are competitive, probabilistic methods provide better calibration along with better uncertainty quantification as we stated in Section 2. There is no aggregation rule that consistently outperforms the other rules in both IID and non-IID partitions for 10 clients.

Table 1: Our main results of 10 clients experiment with means $\pm$ standard errors of the scores across five repetitions/seeds for FMNIST, Cifar-10, and SVHN datasets. Best performing models that overlap within a standard error are highlighted in bold.

| Part. | Model | Agg. | FMNIST | | | Cifar-10 | | | SVHN | | |
|---|---|---|---|---|---|---|---|---|---|---|---|
| | | | Acc(%) ↑ | ECE(%) ↓ | NLL ↓ | Acc(%) ↑ | ECE(%) ↓ | NLL ↓ | Acc(%) ↑ | ECE(%) ↓ | NLL ↓ |
| IID | FED | N/A | $89.62_{\pm0.15}$ | $7.99_{\pm0.12}$ | $0.64_{\pm0.01}$ | $70.09_{\pm0.56}$ | $3.35_{\pm0.07}$ | $0.87_{\pm0.02}$ | $88.55_{\pm0.22}$ | $9.05_{\pm0.17}$ | $0.94_{\pm0.02}$ |
| | FEDAVG | N/A | $89.58_{\pm0.15}$ | $7.97_{\pm0.15}$ | $0.63_{\pm0.01}$ | $70.10_{\pm0.70}$ | $3.31_{\pm0.09}$ | $0.86_{\pm0.02}$ | $88.58_{\pm0.20}$ | $8.97_{\pm0.14}$ | $0.93_{\pm0.01}$ |
| | FVBA | EAA | $\mathbf{90.10_{\pm0.11}}$ | $\mathbf{3.04_{\pm0.09}}$ | $\mathbf{0.30_{\pm0.00}}$ | $67.25_{\pm0.36}$ | $6.76_{\pm0.12}$ | $0.95_{\pm0.01}$ | $\mathbf{90.44_{\pm0.16}}$ | $\mathbf{1.54_{\pm0.05}}$ | $\mathbf{0.36_{\pm0.00}}$ |
| | | GAA | $89.82_{\pm0.11}$ | $6.38_{\pm0.10}$ | $0.46_{\pm0.01}$ | $\mathbf{71.29_{\pm0.21}}$ | $3.11_{\pm0.07}$ | $\mathbf{0.83_{\pm0.01}}$ | $89.58_{\pm0.15}$ | $5.86_{\pm0.08}$ | $0.60_{\pm0.01}$ |
| | | AALV | $89.85_{\pm0.08}$ | $6.44_{\pm0.07}$ | $0.47_{\pm0.00}$ | $70.89_{\pm0.33}$ | $3.15_{\pm0.13}$ | $\mathbf{0.83_{\pm0.01}}$ | $89.53_{\pm0.14}$ | $6.00_{\pm0.08}$ | $0.61_{\pm0.01}$ |
| | | PPA | $89.78_{\pm0.08}$ | $\mathbf{2.16_{\pm0.07}}$ | $\mathbf{0.29_{\pm0.00}}$ | $64.80_{\pm0.31}$ | $10.16_{\pm0.18}$ | $1.05_{\pm0.01}$ | $89.45_{\pm0.16}$ | $4.29_{\pm0.11}$ | $\mathbf{0.38_{\pm0.00}}$ |
| | | CF | $89.90_{\pm0.15}$ | $6.35_{\pm0.10}$ | $0.47_{\pm0.00}$ | $\mathbf{71.17_{\pm0.24}}$ | $\mathbf{2.81_{\pm0.05}}$ | $\mathbf{0.83_{\pm0.01}}$ | $89.38_{\pm0.18}$ | $6.14_{\pm0.15}$ | $0.63_{\pm0.02}$ |
| | FVBWA | EAA | $\mathbf{90.09_{\pm0.05}}$ | $3.07_{\pm0.11}$ | $\mathbf{0.30_{\pm0.00}}$ | $67.32_{\pm0.26}$ | $6.87_{\pm0.20}$ | $0.95_{\pm0.01}$ | $\mathbf{90.51_{\pm0.13}}$ | $1.60_{\pm0.03}$ | $\mathbf{0.36_{\pm0.00}}$ |
| | | GAA | $89.85_{\pm0.07}$ | $6.37_{\pm0.06}$ | $0.47_{\pm0.00}$ | $\mathbf{71.08_{\pm0.35}}$ | $\mathbf{2.86_{\pm0.06}}$ | $\mathbf{0.83_{\pm0.01}}$ | $89.48_{\pm0.14}$ | $5.92_{\pm0.10}$ | $0.61_{\pm0.01}$ |
| | | AALV | $89.85_{\pm0.11}$ | $6.37_{\pm0.06}$ | $0.46_{\pm0.01}$ | $71.05_{\pm0.33}$ | $2.91_{\pm0.03}$ | $\mathbf{0.83_{\pm0.01}}$ | $89.73_{\pm0.13}$ | $5.88_{\pm0.06}$ | $0.60_{\pm0.00}$ |
| | | PPA | $89.68_{\pm0.08}$ | $\mathbf{2.17_{\pm0.09}}$ | $\mathbf{0.29_{\pm0.00}}$ | $65.00_{\pm0.21}$ | $10.00_{\pm0.26}$ | $1.04_{\pm0.00}$ | $89.31_{\pm0.22}$ | $4.34_{\pm0.11}$ | $\mathbf{0.38_{\pm0.00}}$ |
| | | CF | $89.84_{\pm0.12}$ | $6.37_{\pm0.14}$ | $0.46_{\pm0.01}$ | $70.99_{\pm0.30}$ | $3.01_{\pm0.07}$ | $\mathbf{0.83_{\pm0.01}}$ | $89.73_{\pm0.15}$ | $5.91_{\pm0.07}$ | $0.61_{\pm0.00}$ |
| Non-IID | FED | N/A | $88.11_{\pm0.24}$ | $8.73_{\pm0.17}$ | $0.65_{\pm0.01}$ | $65.90_{\pm0.16}$ | $4.39_{\pm0.26}$ | $0.98_{\pm0.00}$ | $86.65_{\pm0.30}$ | $9.88_{\pm0.21}$ | $0.93_{\pm0.01}$ |
| | FEDAVG | N/A | $87.99_{\pm0.35}$ | $8.90_{\pm0.24}$ | $0.66_{\pm0.02}$ | $66.02_{\pm0.29}$ | $4.89_{\pm0.47}$ | $0.97_{\pm0.01}$ | $86.67_{\pm0.38}$ | $9.85_{\pm0.28}$ | $0.94_{\pm0.02}$ |
| | FVBA | EAA | $\mathbf{88.57_{\pm0.21}}$ | $3.45_{\pm0.15}$ | $\mathbf{0.34_{\pm0.01}}$ | $61.48_{\pm0.33}$ | $6.45_{\pm0.87}$ | $1.09_{\pm0.01}$ | $\mathbf{88.33_{\pm0.36}}$ | $\mathbf{1.69_{\pm0.03}}$ | $\mathbf{0.42_{\pm0.01}}$ |
| | | GAA | $\mathbf{88.62_{\pm0.27}}$ | $6.47_{\pm0.20}$ | $0.47_{\pm0.01}$ | $\mathbf{67.26_{\pm0.23}}$ | $3.21_{\pm0.10}$ | $\mathbf{0.93_{\pm0.01}}$ | $87.52_{\pm0.31}$ | $6.24_{\pm0.18}$ | $0.62_{\pm0.01}$ |
| | | AALV | $88.47_{\pm0.20}$ | $6.64_{\pm0.13}$ | $0.47_{\pm0.01}$ | $\mathbf{67.61_{\pm0.51}}$ | $\mathbf{3.21_{\pm0.17}}$ | $0.92_{\pm0.01}$ | $87.66_{\pm0.28}$ | $6.20_{\pm0.16}$ | $0.63_{\pm0.01}$ |
| | | PPA | $88.14_{\pm0.29}$ | $\mathbf{2.41_{\pm0.14}}$ | $\mathbf{0.34_{\pm0.01}}$ | $58.45_{\pm0.48}$ | $13.35_{\pm0.91}$ | $1.26_{\pm0.02}$ | $86.97_{\pm0.21}$ | $7.72_{\pm0.50}$ | $0.48_{\pm0.01}$ |
| | | CF | $\mathbf{88.75_{\pm0.25}}$ | $6.44_{\pm0.11}$ | $0.47_{\pm0.01}$ | $66.99_{\pm0.50}$ | $3.45_{\pm0.09}$ | $\mathbf{0.93_{\pm0.01}}$ | $87.60_{\pm0.29}$ | $6.28_{\pm0.21}$ | $0.63_{\pm0.01}$ |
| | FVBWA | EAA | $88.30_{\pm0.32}$ | $3.47_{\pm0.35}$ | $\mathbf{0.35_{\pm0.01}}$ | $61.01_{\pm1.12}$ | $6.56_{\pm0.71}$ | $1.11_{\pm0.02}$ | $87.91_{\pm0.42}$ | $1.84_{\pm0.11}$ | $\mathbf{0.43_{\pm0.01}}$ |
| | | GAA | $88.39_{\pm0.26}$ | $6.73_{\pm0.22}$ | $0.48_{\pm0.01}$ | $66.07_{\pm0.71}$ | $3.40_{\pm0.27}$ | $0.95_{\pm0.02}$ | $87.56_{\pm0.32}$ | $6.20_{\pm0.17}$ | $0.63_{\pm0.01}$ |
| | | AALV | $88.46_{\pm0.26}$ | $6.78_{\pm0.14}$ | $0.49_{\pm0.01}$ | $67.05_{\pm0.47}$ | $\mathbf{3.03_{\pm0.24}}$ | $0.94_{\pm0.02}$ | $87.44_{\pm0.31}$ | $6.41_{\pm0.16}$ | $0.63_{\pm0.02}$ |
| | | PPA | $87.72_{\pm0.30}$ | $\mathbf{2.47_{\pm0.18}}$ | $\mathbf{0.35_{\pm0.01}}$ | $56.08_{\pm1.78}$ | $12.36_{\pm0.80}$ | $1.32_{\pm0.05}$ | $86.71_{\pm0.27}$ | $8.08_{\pm0.47}$ | $0.50_{\pm0.01}$ |
| | | CF | $88.33_{\pm0.28}$ | $6.90_{\pm0.23}$ | $0.49_{\pm0.01}$ | $66.55_{\pm0.48}$ | $3.33_{\pm0.06}$ | $0.95_{\pm0.01}$ | $87.37_{\pm0.32}$ | $6.42_{\pm0.22}$ | $0.64_{\pm0.02}$ |

**Experiment results with 100 clients** are given in Appendix 6.1 Table 3. VBNNs with a relatively high aggregation degree of spread (EEA and PPA) cannot provide competitive results. In contrast, a relatively low degree of spread yields high performance in all metrics. Furthermore, in the IID partition, high performing aggregation rules significantly surpass the deterministic counterparts. We also measure computational wall clock time per communication round (TPC) in Appendix 6.1.

**Empirical degree of spread.** In Appendix 6.1 Table 4, we report the final models' learned standard deviations. The latter are calculated as the Euclidean norm by stacking standard deviations of all neurons in a vector. As we mention throughout the paper, degree of spread (in our case the standard deviation) is an important factor that affects the outcomes of the VBNNs. When the dataset is coarsely partitioned as in the 10-client experiment, aggregation methods are able to work even though standard deviations are significantly higher for e.g. EAA in FMNIST and SVHN datasets. On the other hand, when the dataset is distributed to a high number of clients as in the 100-client experiment, the aggregations end up with high standard deviations leading to EAA, PPA performing poorly.

## 5 Conclusion

**Summary.** We investigate different distribution aggregation methods for variational Bayesian neural networks in federated learning. First, we derive the variational version of FEDAVG with the Bayesian paradigm, then we benchmark distribution aggregation schemes on three image classification datasets. We compare the performance of the aggregation methods based on prediction accuracy, calibration error, and uncertainty quantification. In general, we observe that spread of distributions highly affects the learning outcomes. We also share our easy-to-use multi-process experimental pipeline providing shorter simulation runtimes.

**Broad Impact.** Our work signifies the importance of aggregation rules for federated learning in a variational Bayesian setting. Furthermore, our findings indicate that aggregated distributions' degree of spread is a notable factor in the federated learning procedure. Our investigations can be further analyzed via the provided implementation and used in different federated learning scenarios including safety-critical domains.

**Limitations and Ethical Concerns.** We present different aggregation schemes for VBNNs with empirical results. Their convergence analysis should be analytically investigated. Furthermore, our

study concerns only the aggregation of Gaussian distributions due to their convenient manipulation and high performance; we consider generic distributions as our future work. The presented aggregations are general-purpose statistical methods, yet they are not investigated from the perspective of fairness sensitive applications. A fairness analysis and testing is required before putting these techniques to use in the field.

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

# 6 Appendix

## 6.1 Experimental Evaluation

In order to validate the proposed aggregation methods via quantitative evaluations, we conduct a series of experiments with VBNNs considering image classification datasets. In this section, we describe baselines, architectures, evaluation metrics, datasets, experiments, training procedure and hyper-parameters.

**Baselines.** We adopt two models to investigate aggregation for VBNN methods namely Federated Variational Bayesian Averaging (FVBA) and Federated Variational Bayesian Weighted Averaging (FVBWA). As counterparts for VBNNs, we use FEDAVG and FED to create deterministic baselines. In FEDAVG and FVBWA, the aggregation weights are equal to $\beta_k = \frac{|\mathcal{D}_k|}{|\mathcal{D}|}$ where $|\mathcal{D}_k|$ denotes the cardinality of the $k^{th}$ dataset. In FVBA and FED, we assume that each client contains the same amount of information for the aggregation regardless of their data; therefore, we set $\beta_k = \frac{1}{K}$.

**Architectures.** We follow the same experimental pipeline and architectures from [19]. We use two different architecture for NNs and VBNNs. Both networks have two $5 \times 5$ convolutional layers. ReLU activation function and $2 \times 2$ max-pooling operation follows the convolutional layers. For NNs, there are three linear layers with $400$, $120$, and $84$ hidden dimensions respectively. Between dense layers, we apply ReLU activation function. For VBNNs, we replace linear layers with variational Bayesian linear layers which are parameterized with mean and log variance. Instead of learning point estimate weights, variational Bayesian layers consist of $\mu$ and $\sigma^2$ for each neuron; thus, variational Bayesian layers have twice the number of parameters than dense layers.

**Metrics.** We evaluate the performance of the models' predictions using Accuracy (Acc), Expected Calibration Error (ECE) as a measure of prediction calibration [25], and Negative Log-Likelihood (NLL) as a measure of model fit that quantifies the uncertainty. For calculate ECE, we create bins for instances and calculate $ECE = \sum_{m=1}^{M}(B_m/M)(a_i - c_i)$ where $M$ is the count of the bin, $B_m$ is the count of instance located in that bin, $a_i, c_i$ are accuracy and average confidence in that bin respectively. For a fair comparison with other works, we report the scores of the models after the final communication round.

**Datasets.** We benchmark models and aggregation rules in three image classification datasets that are FMNIST [32], Cifar-10 [17], and SVHN [26]. The details of the datasets are listed in Table 2.

**Experiments.** To demonstrate the performances of aggregation rules, we exhibit two different experiment setups that are based on active client numbers with two different data distributions on three real-world image classification datasets. In the first setup, we simulate a scenario where all the clients are active in all communication rounds. However, this is not always possible in real-life scenarios; therefore,

Table 2: Details on datasets. Image sizes are given as Channel $\times$ Height $\times$ Width.

| Datasets | Image Size | #Labels | Train Size | Test Size |
|---|---|---|---|---|
| FMNIST | $1 \times 28 \times 28$ | | 60000 | 10000 |
| Cifar-10 | $3 \times 32 \times 32$ | 10 | 50000 | 10000 |
| SVHN | $3 \times 32 \times 32$ | | 73257 | 26032 |

we imitate the inactivity of the clients in the second setup. In both setups, there are always 10 clients that are active for communication, however, in the second one, the active clients are randomly sampled from 100 clients. Besides the number of clients, another important difference between setups is the dataset size since the whole dataset is divided into 100 clients. To simulate the data distribution shift of the clients, we conduct the experiments with two different data partitioning techniques like in [19]. The first partition is named IID where each client has approximately the same amount of data for each label; while, in the second partition, non-IID, clients' data distribution is generated with a Dirichlet distribution in a way that each client has a different amount of data for each label.

**Training procedure and hyper-parameters.** For each client optimization, we use Stochastic Gradient Descent (SGD) with the parameters $0.01$ for learning rate, $0.9$ for momentum, and $10^{-5}$ for weight decay. We employ Cross Entropy loss for deterministic models (FED and FEDAVG) and negative variational free energy loss in Eq. 3 for probabilistic models (FVBA, FVBWA). For VBNNs,

Table 3: Our main results of 100 clients experiment with means $\pm$ standard errors of the scores across five repetitions for FMNIST, Cifar-10, and SVHN datasets. Best performing models that overlap within a standard error are highlighted in bold.

| Part. | Model | Agg. | FMNIST | | | Cifar-10 | | | SVHN | | |
|---|---|---|---|---|---|---|---|---|---|---|---|
| | | | Acc(%) ↑ | ECE(%) ↓ | NLL ↓ | Acc(%) ↑ | ECE(%) ↓ | NLL ↓ | Acc(%) ↑ | ECE(%) ↓ | NLL ↓ |
| IID | FED | N/A | 88.55±0.15 | 8.30±0.08 | 0.63±0.01 | 64.40±0.51 | 12.96±0.21 | 1.16±0.02 | 87.20±0.14 | 9.77±0.11 | 0.99±0.01 |
| | FEDAVG | N/A | 88.38±0.12 | 8.48±0.08 | 0.64±0.02 | 64.35±0.70 | 12.97±0.30 | 1.17±0.02 | 87.08±0.14 | 9.91±0.07 | 1.01±0.02 |
| | FVBA | EAA | 85.77±0.11 | 3.02±0.29 | 0.42±0.00 | 45.10±0.32 | 13.27±0.39 | 1.56±0.01 | 79.51±0.57 | 12.34±0.57 | 0.74±0.02 |
| | | GAA | **89.49**±0.17 | 4.97±0.11 | **0.35**±0.00 | 66.97±0.37 | **3.52**±0.22 | **0.94**±0.01 | **90.16**±0.06 | **2.93**±0.03 | **0.40**±0.00 |
| | | AALV | 89.31±0.17 | 5.03±0.12 | **0.35**±0.00 | **67.45**±0.47 | **3.53**±0.21 | **0.93**±0.01 | **90.10**±0.12 | 3.08±0.07 | **0.40**±0.01 |
| | | PPA | 85.05±0.09 | 3.24±0.14 | 0.45±0.00 | 43.47±1.11 | 12.03±0.54 | 1.60±0.02 | 70.36±1.24 | 18.06±0.84 | 1.08±0.04 |
| | | CF | 89.40±0.11 | 4.96±0.06 | **0.35**±0.00 | 67.27±0.71 | 3.65±0.29 | 0.95±0.02 | 89.94±0.14 | 3.16±0.07 | **0.40**±0.00 |
| | FVBWA | EAA | 85.88±0.14 | **2.58**±0.15 | 0.42±0.00 | 44.99±0.83 | 12.36±0.60 | 1.55±0.02 | 80.52±0.22 | 11.74±0.50 | 0.71±0.01 |
| | | GAA | **89.56**±0.13 | 4.81±0.11 | **0.35**±0.00 | 67.44±0.67 | 3.63±0.24 | **0.93**±0.02 | **90.13**±0.15 | 2.97±0.08 | **0.40**±0.01 |
| | | AALV | **89.48**±0.12 | 4.92±0.12 | **0.35**±0.00 | 66.48±0.49 | 4.10±0.17 | 0.96±0.01 | **90.13**±0.10 | 3.11±0.02 | **0.40**±0.00 |
| | | PPA | 85.00±0.13 | 3.14±0.30 | 0.45±0.00 | 42.76±0.81 | 11.84±0.28 | 1.61±0.02 | 66.14±3.41 | 18.07±1.09 | 1.20±0.09 |
| | | CF | **89.53**±0.11 | 4.87±0.05 | **0.35**±0.00 | **67.09**±0.20 | 3.96±0.29 | 0.95±0.01 | 90.02±0.08 | 3.18±0.06 | **0.40**±0.01 |
| Non-IID | FED | N/A | 87.16±0.13 | 8.39±0.15 | 0.58±0.01 | **61.23**±1.00 | 10.39±1.18 | 1.16±0.04 | 85.00±0.61 | 10.28±0.38 | 0.91±0.03 |
| | FEDAVG | N/A | 86.77±0.23 | 8.56±0.34 | 0.59±0.02 | **60.26**±1.12 | 10.97±1.38 | 1.19±0.04 | 84.82±0.80 | 10.18±0.48 | 0.91±0.04 |
| | FVBA | EAA | 82.18±0.78 | **4.32**±0.60 | 0.54±0.02 | 26.87±4.52 | 5.59±0.67 | 1.93±0.11 | 67.47±1.80 | 12.90±0.80 | 1.08±0.06 |
| | | GAA | **87.73**±0.14 | 4.72±0.21 | **0.38**±0.01 | **60.85**±1.45 | 4.22±1.04 | **1.10**±0.04 | **87.34**±0.35 | 2.75±0.32 | **0.46**±0.01 |
| | | AALV | 87.49±0.29 | 4.94±0.25 | **0.38**±0.01 | **60.60**±1.72 | 4.84±1.61 | 1.11±0.05 | **87.46**±0.35 | 2.76±0.25 | **0.46**±0.01 |
| | | PPA | 72.87±4.12 | 7.97±1.05 | 0.83±0.13 | 19.06±2.33 | 4.92±0.82 | 2.13±0.06 | 18.28±0.81 | 4.36±1.60 | 2.25±0.02 |
| | | CF | **87.63**±0.22 | 4.75±0.17 | **0.38**±0.01 | **60.79**±1.60 | 4.36±1.39 | **1.10**±0.05 | **87.12**±0.25 | **2.99**±0.31 | 0.47±0.01 |
| | FVBWA | EAA | 79.61±1.06 | 4.74±0.80 | 0.61±0.02 | 25.29±3.98 | 5.38±0.80 | 1.97±0.11 | 52.29±8.42 | 13.68±1.95 | 1.51±0.24 |
| | | GAA | 87.40±0.26 | 4.92±0.24 | **0.39**±0.01 | 59.56±1.75 | 5.22±1.42 | 1.13±0.05 | **87.18**±0.22 | 2.91±0.35 | 0.47±0.01 |
| | | AALV | 87.42±0.40 | 5.07±0.30 | **0.39**±0.01 | 58.99±1.83 | 5.81±1.76 | 1.14±0.06 | 86.84±0.47 | 3.35±0.41 | 0.48±0.02 |
| | | PPA | 76.38±0.79 | 5.59±0.69 | 0.70±0.02 | 18.55±2.22 | **3.37**±0.38 | 2.15±0.05 | 21.22±2.81 | 7.37±2.75 | 2.22±0.04 |
| | | CF | 87.16±0.41 | 5.12±0.36 | 0.40±0.02 | 59.44±1.75 | 5.86±1.95 | 1.14±0.06 | **87.23**±0.45 | 3.07±0.46 | **0.47**±0.02 |

we select the standard normal distribution $\mathcal{N}(0,1)$ as the prior distribution $p(\theta)$ like in [15]. We set $E = 10$ which is the number of local epochs. Following the benchmark paper [19], we conduct our experiments with $T = 50$ epochs for 10 clients and $T = 500$ for 100 clients. We benchmark the models with 5 different seeds from 0 to 4 on Intel Xeon CPU E5-2690 v3 and Nvidia Quadro P6000 24 GB.

**Implementation and reproducibility.** We provide a PyTorch implementation of our experiment pipeline. In order to simulate the federated learning approach, we execute the clients in parallel with multi-processing which significantly accelerates the training procedure. The code is available at `https://github.com/ituvisionlab/BFL-P`. We also share the seeds that are used in the experiments, hence our data splits (IID; non-IID), all initializations, and the presented results are reproducible.

**Runtime comparison.** We measure computational wall clock time per communication round (TPC) as in Table 5. Comparing the single process case versus others, our multi-process pipeline significantly speeds up the computational time cost for the simulation. Furthermore, there is no significant TPC difference among aggregation methods except PPA due to the sampling process.

Table 4: Comparison of standard deviations (as means $\pm$ standard errors) across five repetitions for FMNIST, Cifar-10, and SVHN datasets. The standard deviations belong to the settings with Non-IID partitioned datasets, FVBWA baselines, and learned final models. The lowest three rank standard deviations are highlighted in bold.

| Exp. | Agg. | FMNIST | Cifar-10 | SVHN |
|---|---|---|---|---|
| | | Standard Deviation | | |
| 10 client | EAA | 15.48±0.41 | 17.65±0.29 | 17.90±0.38 |
| | GAA | **3.93**±0.01 | **4.68**±0.00 | **4.67**±0.01 |
| | AALV | **2.81**±0.01 | **3.22**±0.00 | **3.48**±0.01 |
| | PPA | 18.40±0.11 | 23.83±0.17 | 31.62±0.38 |
| | CF | **1.40**±0.02 | **1.65**±0.04 | **1.57**±0.02 |
| 100 client | EAA | 79.19±0.66 | 91.79±1.16 | 94.54±0.80 |
| | GAA | **3.94**±0.01 | **4.70**±0.01 | **4.69**±0.01 |
| | AALV | **2.80**±0.00 | **3.22**±0.00 | **3.49**±0.00 |
| | PPA | 64.70±0.26 | 75.38±0.53 | 101.73±0.61 |
| | CF | **1.42**±0.03 | **1.75**±0.05 | **1.70**±0.06 |

Table 5: Runtime comparison results based on the number of processes of IID partitioned 100 clients experiment with means $\pm$ standard errors of Time Per Communication round (TPC) across five communication rounds for CIFAR-10 dataset. Multi-processed pipeline with 10 processes is the fastest for all models.

| Model | Agg. | 1 process | 5 processes | 10 processes |
|---|---|---|---|---|
| | | **Time per Communication Round** | | |
| FED | N/A | $60.83_{\pm0.26}$ | $15.55_{\pm0.51}$ | $9.30_{\pm0.07}$ |
| FEDAVG | N/A | $60.90_{\pm0.18}$ | $15.57_{\pm0.39}$ | $9.22_{\pm0.20}$ |
| FVBA | EAA | $72.77_{\pm0.18}$ | $16.22_{\pm0.05}$ | $9.49_{\pm0.06}$ |
| | GAA | $71.23_{\pm0.88}$ | $16.48_{\pm0.10}$ | $9.41_{\pm0.05}$ |
| | AALV | $72.10_{\pm0.36}$ | $16.33_{\pm0.10}$ | $9.51_{\pm0.09}$ |
| | PPA | $66.95_{\pm0.31}$ | $18.06_{\pm0.20}$ | $11.23_{\pm0.16}$ |
| | CF | $72.53_{\pm0.29}$ | $16.34_{\pm0.10}$ | $9.36_{\pm0.14}$ |
| FVBWA | EAA | $72.38_{\pm0.31}$ | $16.45_{\pm0.06}$ | $9.44_{\pm0.11}$ |
| | GAA | $72.78_{\pm0.15}$ | $15.88_{\pm0.25}$ | $9.42_{\pm0.11}$ |
| | AALV | $72.41_{\pm0.24}$ | $16.19_{\pm0.09}$ | $9.64_{\pm0.13}$ |
| | PPA | $67.51_{\pm0.16}$ | $17.99_{\pm0.42}$ | $11.15_{\pm0.12}$ |
| | CF | $72.86_{\pm0.40}$ | $17.22_{\pm0.40}$ | $10.56_{\pm0.08}$ |

