# OpenReview forum: "How to Combine Variational Bayesian Networks in Federated Learning"
_NeurIPS.cc/2022/Workshop/Federated_Learning — FL-NeurIPS 2022 Poster_

### Official Review · Reviewer_HRSZ · 2022-10-10
**Empirical comparison of five different aggregation schemes for Variational Bayesian Neural Networks**

This paper empirically compares five different statistical aggregation schemes for Variational Bayesian Neural Networks on image classification benchmarks.

While current FL methods just average model or gradient parameters, FL on VBNN requires an understanding of how to best average _parametrized distributions_  on the server. This work advances this understanding, although the presented numerical results are not very conclusive yet. However, I think this work could still be of interest to part of the community.

A few comments:
- The illustrations is Section 3 explain well the different aggregation schemes. However, in the first pass I got the impression that the work only focuses on one-dimensional Gaussian distributions. I think the connection to the implemented scheme could be made more clear at the beginning of the section.
- The experiments use the same hyperparameters (e.g. stepsize) for all aggregation schemes. It would be great if this could be further justified.

---

### Official Review · Reviewer_V3ee · 2022-10-14
**Interesting empirical study with some open questions**

Thanks for that study and write-up, I read it with great curiosity and as often, it left me with a bunch more questions, some of which you might be able to answer in this paper.
The writing quality and clarity is high, it is original as claimed by the authors and the significance is to be seen in slightly larger-scale and more challenging scenarios.

My biggest open question from the theoretical side of things concerns the graphical model underlying the different aggregation strategies.
From the perspective of e.g. obtaining p(\theta|D), i.e. the posterior (or approximation) of the global data-set, which is the `correct` aggregation method? - is there even one?
FedSparse for example makes a clear assumption on how the local prior is determined and how aggregation on the server corresponds to the M-step in EM under the specific graphical model.
How can we justify any alternative methods from the perspective of bayesian inference? Ideally, the posterior, especially the uncertainty over parameters should be determined by choice of graphical model and the data, not by ad-hoc choice of aggregation strategy.

In Line 117, are you referring to the posterior $p(\theta_k|D_k)$ being approximated by q?

---

### Official Review · Reviewer_uZ1p · 2022-10-19
**The paper proposes multiple probabilistic aggregation schemes for Bayesian neural networks in the context of federated learning**

Overall, the paper is well written. The paper deals with the scenario where the neural network parameters in each of the client's model is not a point estimate, rather they have distributions. Authors have used variational inference in a local way to learn the client NN distributions and then experimented with different aggregation strategies at server to check for performance regarding the global optimization.

---

### Decision · Program_Chairs · 2022-10-20

Accept (Poster)